# Implications of the prevalence and magnitude of sustained declines for determining a minimum threshold for favourable population size

Rhys E. Green[1,2], Gillian Gilbert[3]*, Jeremy D. Wilson[4], Kate Jennings[5]

1 Department of Zoology, University of Cambridge, Cambridge, United kingdom, 2 RSPB Centre for Conservation Science, Sandy, Bedfordshire, United kingdom, 3 RSPB Centre for Conservation Science, Glasgow, United kingdom, 4 RSPB Centre for Conservation Science, Edinburgh, United kingdom, 5 RSPB Department of Site Conservation Policy, Sandy, Bedfordshire, United kingdom

* Gillian.Gilbert@rspb.org.uk

**Data Availability Statement:** All relevant data are within the manuscript and its Supporting Information files S1 Table.

## Abstract

We propose a new approach to quantifying a minimum threshold value for the size of an animal population, below which that population might be categorised as having unfavourable status. Under European Union law, the concept of Favourable Conservation Status requires assessment of populations as having favourable or unfavourable status, but quantitative methods for such assessments have not yet been developed. One population threshold that is well established in conservation biology is the minimum viable population (MVP) defined as the size of a small but stable population with an acceptably low risk of extinction within a specified period. Our approach combines this small-population paradigm MVP concept with a multiplier, which is a factor by which the MVP is multiplied to allow for the risk of a sustained future decline. We demonstrate this approach using data on UK breeding bird population sizes. We used 43-year time-series data for 189 species and a qualitative assessment of population trends over almost 200 years for 229 species to examine the prevalence, duration and magnitude of sustained population declines. Our study addressed the problem of underestimation of the duration and magnitude of declines caused by short runs of monitoring data by allowing for the truncation of time series. The multiplier was derived from probability distributions of decline magnitudes within a given period, adjusted for truncation. Over a surveillance period of 100 years, we estimated that there was a 10% risk across species that a sustained population decline of at least sixteen-fold would begin. We therefore suggest that, in this case, a factor of 16 could be used as the multiplier of small-population MVPs to obtain minimum threshold population sizes for favourable status. We propose this 'MVP Multiplier' method as a new and robust approach to obtaining minimum threshold population sizes which integrates the concepts of small-population and declining-population paradigms. The minimum threshold value we propose is intended for use alongside a range of other measures to enable overall assessments of favourable conservation status.

**Funding:** This study was funded by the Royal Society for the Protection of Birds, UK.

**Competing interests:** The authors have declared that no competing interests exist.

## Introduction

Biodiversity is under unprecedented global pressure from anthropogenic change. Declines, extirpations and extinctions of populations of wild species are occurring widely [1,2]. Given that conservation resources are limited, efficient quantitative methods are needed to decide whether a species' population and habitats are in a healthy state, and what the desired outcome of conservation efforts should be [3,4]. National and international environmental laws may require assessments of status of populations of wild species as being, for example, 'favourable' [5, 6] or 'recovered' [7, 8, 9]. For example, the Endangered Species Act (ESA) in the United States [7], the Habitats [10], and Birds [11] Directives in the European Union and the Bonn Convention [12] globally, require such assessments. Such assessments can also provide a rationale for the prioritisation of conservation action [7,8,10,11]. However, these terms tend not to have consistent quantitative definitions. In our view, there is not yet any adequate, quantitative method to calculate a threshold for favourable population size to contribute to wider assessments of species' conservation status. It has been suggested that a favourable population might be attained when the population of a species in a region is at or close to its carrying capacity under land cover conditions which are ideal for it [5, 13, 14]. However, this is not a workable approach because it ignores trade-offs among species with contrasting requirements in any fixed land area. Under this potential carrying capacity approach, actions required to deliver a favourable population outcome for one set of species are likely to make conditions unfavourable for others with contrasting habitat requirements. We do not see a rational way to reconcile these conflicts among the contrasting ideal requirements of species. Additional difficulties in using the carrying capacity approach for decisions on favourable status have been considered by Trouwborst *et al.* [15].

Another approach is to define acceptable population targets by reference to past populations. For example, in the U.S. and Canada, numerical population targets are set for all bird species of continental and regional importance within the Partners in Flight Landbird Conservation Plans [16]. The targets are set to reinstate what the national population would have been 30 years ago, with the baseline population estimates and trends being derived from distribution and abundance data [16]. This approach is pragmatic, but susceptible to the shifting baseline syndrome, in which what is deemed acceptable is defined by reference to a purely arbitrary status in the recent past, when species' populations may already have been depleted [17]. Because it uses measurements of declines over a relatively short timescale, this approach does not account for declines occurring over longer periods.

Threshold measures of favourable population size, which do not involve ideal carrying capacity or arbitrary baselines in the recent past, may use estimates of minimum viable population size (MVP). This approach involves maintaining population size at a level which keeps the risk of population extinction below an acceptable threshold. If data on density dependence and the effects of environmental and demographic stochasticity on demographic rates, population growth rate and carrying capacity are available over a sufficiently long interval, a population viability analysis can be performed to estimate the long-term probability of extinction. However, such detailed and long-term datasets are extremely rare. More often, there are sufficient data to model the probability of extinction based upon short runs of data for periods when population size shows little or no consistent trend. This situation is described by the small-population paradigm of Caughley [18], in which a minimum threshold population size can be defined, above which there would be an acceptably low risk of extinction within a defined period. Under this paradigm, a population above the MVP, with mean demographic rates sufficient to allow it to be stable, would probably persist in the long-term, whereas one below the MVP with the same mean demographic rates would be unacceptably liable to

decline to extinction because of environmental and demographic stochasticity. However, models fitted to such short runs of data are likely to underestimate the effects of sustained population declines caused by long-term deterioration in environmental conditions. We therefore suggest that the threshold for favourable population size should certainly be greater than this small-population paradigm MVP, because future adverse changes in external factors affecting a small stable population could cause it to undergo a sustained and unforeseen decline. How much greater than the small-population paradigm MVP should that threshold be? Clearly, the answer depends upon the frequency and magnitude of unforeseen sustained population declines in the real world. In this paper, we focus on developing a standardised method for establishing a factor by which a species-specific MVP should be multiplied to protect against unforeseen future declines. To do this, we use data on observed sustained population declines of wild birds in the UK to estimate the value of this factor or multiplier. Although this illustrative example involves birds, we suggest that our approach could also be applied to other taxa for which long-term population data exist, such as the large number of vertebrate species covered by population time series in the Living Planet Index [19]

## Methods

### Implications of truncation in time-series data

Periods over which animal population sizes are measured repeatedly using comparable methods usually span a few decades at best, which is short relative to the duration of some declines. Our main analytical challenge was to develop methods to allow for such data truncation and thus avoid underestimation of the prevalence and magnitude of declines of long duration. We achieved this by calibrating results from detailed, quantitative, but shorter-term trend data (43 years) [20] against data from a qualitative survey of population trends over almost 200 years [21].

### Quantitative data on breeding bird populations in the UK

We used breeding bird population data from the UK, where there is a relatively long history of quantitative bird population monitoring. The dataset we used was originally compiled to assess the conservation status of all UK breeding bird species [20] and covers the period 1970 to 2013. Annual population estimates or population indices believed to be directly proportional to population size were available for many species, but species with occasional national surveys at longer intervals were also included. The results we used were smoothed trends fitted to annual data from the British Trust for Ornithology (BTO) and Joint Nature Conservation Committee (JNCC) Common Birds Census (CBC) and the BTO/JNCC/Royal Society for the Protection of Birds (RSPB) Breeding Bird Survey (BBS) [22], annual data from the Rare Breeding Birds Panel [23], and national population estimates from surveys of single species, which were typically undertaken at intervals of at least several years [20]. The decline data used are in supporting information S1 Table.

We analysed data for 189 species native to the UK. The beginning and end of each time series was taken to be the first $t_{initial}$ and last $t_{last}$ years with a population index or estimate, so that ($t_{last}$—$t_{initial}$) defines the surveillance period. Surveillance periods were 20 years or more for 167 species and covered the whole 43-year period 1970–2013 for 129 species. We then used these data (i) to identify sustained population declines (SPDs) and (ii) to quantify their attributes (duration and magnitude) and the extent to which these estimates were affected by data truncation, and (iii) to then account for the effect of truncation on SPD attributes.

**Identification of sustained population declines.** The surveillance periods were searched for SPDs, which we defined using the following rules. We identified the longest period in the

series, subject to a minimum of 10 years, in which the population or index $n_{start}$ at the beginning of the period $t_{start}$ was larger than the population $n_{stop}$ at the end of the period $t_{stop}$, and with all the intervening population values within the period being smaller than $n_{start}$. We omitted data for two species for which the duration of the time series was fewer than ten years, leaving 187 species for further analysis. If such a period was identified it was considered a candidate SPD and was evaluated against the following additional criteria. Beginning at $t_{start}$, the candidate period was searched until a population size value occurred that was smaller than the next value in the series. This value was termed $n_{low}$ and the year in which it occurred was termed $t_{low}$. If values of $n$ in all the years of the candidate decline period after $t_{low}$ remained above $n_{low}$ the decline was deemed to have ended at year $t_{low}$. The population at the end of the sustained decline $n_{stop}$ was then set at $n_{low}$ and the year of cessation of the decline $t_{stop}$ was taken to be $t_{low}$. Otherwise, the decline was deemed to have continued beyond $t_{low}$. In that case, the process was repeated until conditions were satisfied for the end of the decline or the end of the candidate SPD was reached. Once the end of an SPD was identified, the procedure described above was repeated on all the later years of the time series to identify any further candidate SPDs. All candidate SPDs reaching the arbitrary minimum duration of 10 years were then confirmed as SPDs, with more than one SPD possible for each species.

Some SPDs identified by this method might have begun before $t_{start}$ or ended after $t_{stop}$, but this may not have been detected because the period of surveillance was short. To take this into account, we identified those SPDs for which the start time $t_{start}$ occurred fewer than five years after $t_{initial}$, or ended fewer than five years before $t_{last}$, as having uncertain duration and magnitude, with their calculated magnitudes and durations therefore being minima because of left-censoring of the data, right-censoring or both. Our analyses of SPD duration, taking censoring into account, are analogous to survival analyses with censoring (see Kalbfleisch [24]). In this case, the persistence of an ongoing SPD from one year to the next can be regarded as equivalent to the survival of an individual.

**Attributes of sustained population declines.** We detected a total of 82 SPDs, involving 80 species. For most species there was one SPD, but two SPDs were detected for two species. It was possible to define the start of 43 of the 82 SPDs, with the remaining 39 declines already being in progress at the start of the time series or beginning within five years. For 46 of the 82 SPDs, including 25 of the 43 with defined start dates, the end of the decline was not well-defined because the apparent stop date ($t_{stop}$) was within five years of the end of the time series ($t_{end}$), which we took to indicate that there was insufficient evidence that the decline had stopped. Hence, there were only 18 SPDs (22%) for which both start and stop dates were well-defined, and were thus complete. The remainder were classified as truncated, and hence their apparent durations and magnitudes were minima.

Finally, whether complete or truncated, we defined the magnitude $m$ of an SPD as the factor by which population size was estimated to have declined, $n_{start}/n_{stop}$, using the last year with a population greater than zero as the denominator in cases where the population declined to extinction. The duration of the SPD was taken to be $t_{stop} - t_{start}$.

**Analysis of frequency distribution of SPD durations to account for truncated declines.** Having begun, an SPD might be assumed to have a constant annual probability of ending in any given year. If that was so, it would allow a simple statistical description of the distribution of SPD durations in which the probability density function f($d$) of SPD duration $d$ is modelled as the exponential decay function,

$$f(d) = \left(\frac{1}{\theta}\right) \exp(-\theta\, d),$$

where θ is the annual probability that an ongoing SPD comes to an end because the population becomes stable or starts to increase. If the assumption of exponential decay is an acceptable approximation, it would then be possible to use data on all SPDs to estimate θ, regardless of whether the start date of the SPD was known. We tested this assumption by plotting a Kaplan-Meier graph of the proportion of the 43 declines with known starts for which the decline was still in progress in each successive year. We also estimated the annual probability of a decline stopping under our hypothesis of a fixed annual probability by the maximum-likelihood method of Kalbfleisch [24] for right-censored lifetimes. Both the Kaplan-Meier plot and maximum-likelihood analysis allow for right-censoring caused by truncation at the end of the time series. We used a χ2 test to compare observed and expected numbers of declines in each of three categories of duration (10–13, 14–19 and 20–34 years), defined to avoid expected frequencies per class of fewer than five. The purpose of this was to test the assumption that the cessation rate of SPDs could reasonably be modelled using the exponential decay function. Having found that the exponential model fitted complete SPDs reasonably well (see Results), we used the maximum-likelihood method of Kalbfleisch [24] to estimate θ from all 82 SPDs, including those with uncertain start dates. We estimated the arithmetic mean duration of SPDs as $-1/log_e(1-\theta)$

**Analysis of frequency distribution of SPD magnitudes to account for truncated declines.** We plotted the relationship between SPD magnitude and duration for all 82 SPDs. We considered this acceptable because the mean annual rate of population decline $v$, estimated for a given SPD as

$$v = 1 - m^{(1/(tstop-tstart))}$$

did not differ significantly between complete (N = 18) and truncated (N = 64) SPDs (Mann-Whitney U test, $U = 453$, $P = 0.168$). Neither was the mean annual rate of population decline significantly correlated with decline duration (Spearman rank correlation $r_S = -0.020$, $P = 0.857$). Not surprisingly, long SPDs had larger magnitudes ($m$) than short SPDs ($r_S = 0.491$, $P < 0.0001$). This lack of dependence of $v$ on duration allows us to model decline rates and SPD cessation rates separately and combine the results later to model decline magnitudes.

The values of $v$ appeared to be log-normally distributed and this was tested and supported by estimating the least squares mean and standard deviation of $log_e v$ and comparing the observed and modelled cumulative distributions using a Kolmogorov-Smirnov one-sample test of goodness-of-fit (see Results). We therefore used a model of decline magnitude in which we assumed that $log_e v$ varied among species according to a normal distribution with mean $\mu$ and standard deviation $\sigma$, but that the rate of decline was unrelated to decline duration. Under this model, the expected geometric mean decline magnitude $m$ for SPDs of duration $d$ is given by

$$(1 - \mu)^{-d},$$

and the probability density function of decline magnitude g($m,d$) at a duration $d$ for a set of SPDs which are still in progress is given by the normal distribution of

$$log_e(1-\exp(-log_e(m))^{1/d}),$$

with mean $\mu$ and standard deviation $\sigma$. We obtained the expected probability density of SPDs of all durations by integrating numerically with respect to $d$ the product of f($d$) and g($m,d$) for each value of $m$, using our estimates of the mean and standard deviation of $log_e v$ as $\mu$ and $\sigma$. This procedure gives the unbiased probability density distribution of SPD magnitudes that would be expected if surveillance periods were indefinitely long.

## The effect of duration of the surveillance period on the prevalence of SPDs

We have now used the 1970–2013 time-series data to estimate the frequency distribution of SPD durations and magnitudes, accounting for the effect of data truncation. However, we did not consider that the 1970–2013 time-series data set was sufficient to estimate SPD prevalence (defined as the probability that a species' population will be affected by an SPD during a surveillance period of defined length during which population size was monitored reliably) because of the limited maximum duration of the time series and the fact that surveillance periods varied among species. We therefore also estimated the prevalence of SPDs using results from a review of long-term changes in breeding populations of birds in the UK [21] by Gibbons, Avery & Brown (henceforth termed GAB). GAB assessed trends in breeding populations of all bird species in the UK qualitatively over almost 200 years, between 1800 and 1995. They assigned scores on an eleven-point integer scale to indicate the magnitude and direction of population trends in each of five time periods of varying duration (25–49 years: 1800–1849, 1850–1899, 1900–1939, 1940–1969 and 1970–1995). The scores were defined in terms of rates of population increase or decrease, ranging from "huge decrease" (-5) to "huge increase" (+5), with species whose population showed little or no trend during a period, or fluctuating numbers, being assigned the central value of zero. The GAB scores were based upon previous reviews of historical data by other authors and their own assessment of recent trends. Given the duration of the periods assessed by GAB, the declines they identified are likely to have minimum durations broadly comparable with the ten-year minimum we used in our definition of SPDs based upon 1970–2013 quantitative data.

We used data for 229 native species (from Table 2 within GAB [21]) and calculated the proportion of species $p$ for which a decline (score -1 to -5) was recorded for each of the six GAB time periods. We then combined GAB results for pairs of consecutive periods and calculated the proportion of species in which a decline was recorded in either or both of the component shorter periods within the composite period. We repeated this for all possible sets of composite periods comprising three, four, five and six consecutive GAB periods. This gave us a set of 15 surveillance periods with durations, $s$, varying from 25 years to 195 years (1800–1995) and estimated proportions, $p$, of species with a decline in one or more of the component periods. Having inspected a plot of $p$ against $s$ to assess a plausible shape for the relationship, we fitted the least squares regression of $\log_e(1-p)$ on $s$ and estimated $p$ for a given $s$ as

$$p = 1 - \exp(b_0 + b_1 s),$$

where $b_0$ and $b_1$ are the fitted intercept and slope of the regression. We assessed whether the GAB analyses gave an approximation of SPD prevalence that was comparable with our more recent short-term quantitative results, by comparing our single estimate of SPD prevalence for 1970–2013 with that for the overlapping GAB period (1970–1995) and the value predicted by the regression model.

## Combining SPD magnitude and prevalence data to estimate the exceedance distribution of SPD magnitudes beginning in a surveillance period of defined length

We were now able to integrate our analyses of SPD prevalence, duration and magnitude to estimate the proportion of all species that would be subject to an SPD of a given magnitude beginning in a surveillance period of a specified duration (100 years). This procedure covered the whole range of magnitudes, from zero upwards. To do this, we converted the calculated probability density distribution of SPDs to an exceedance (negative cumulative) probability distribution and then multiplied the exceedance probabilities for each decline magnitude by

the proportion of species expected to undergo a population decline within a surveillance period of defined length. We used the regression results described above to estimate the proportion of species with a decline in a surveillance period of 100 years.

We calculated 95% confidence limits of all results using a non-parametric bootstrap method with species as the unit for bootstrap resampling. We resampled the data for $N$ species by drawing a sample of $N$ at random with replacement. We then performed the analyses described above on the bootstrap sample and recorded the estimated parameter values and quantities derived from them. We performed the bootstrap resampling 10,000 times and took the bounds of the central 9,500 bootstrap parameter estimates or derived values to define the 95% confidence limits.

## Results

### Prevalence of SPDs in relation to the duration of the surveillance period

We detected one or more SPDs in 80 of 187 species (43%). Surveillance periods averaged 38.3 years across the 187 species. This is close to the maximum value of 43 years if surveillance of all species had covered the entire period.

Analysis of GAB population trend scores for six 25–49-year periods in the 195-year period 1800–1995 indicated that the proportion $p$ of species in which a decline was recorded in at least one period increased with increasing duration of the composite surveillance periods (Fig 1). The least-squares estimates of the intercept and slope parameters $b_0$ and $b_1$ were -0.3153 and -0.0031 respectively when any negative GAB score was taken to indicate an SPD (Fig 1). A significance test of this relationship would not be appropriate because the observations in the composite periods are not mutually independent. There was close agreement between both the SPD prevalence calculated from the recent quantitative population data for 1970–2013 and the GAB results for the most recent overlapping GAB period (1970–1995), and the prevalence predicted from the GAB regression (Fig 1). We therefore considered that the agreement between the GAB regression and the SPD prevalence for 1970–2013 was sufficiently good for us to use the GAB regression to predict the prevalence of SPDs in any surveillance period up to 200 years. In our case, having selected a surveillance period of 100 years, the predicted prevalence of SPDs from the GAB regression was 0.464 (95% confidence limits 0.413–0.515).

### Duration of SPDs

Kaplan-Meier and maximum-likelihood analysis of SPD durations of UK birds in the period 1970–2013 indicated that an exponential model of the annual probability of cessation of a decline gave a satisfactory fit to the data. The exponential model fitted to the data for the 43 declines with a known start date showed a good agreement between the observed and modelled SPD stop times (Fig 2A). The distribution of stop times of the 18 complete SPDs showed no indication of departure from the distribution expected from the fitted exponential model (goodness-of-fit $\chi^2_{(2)} = 0.253$, $P = 0.881$). The maximum-likelihood estimate of the annual stopping rate parameter $\theta$ from the subset of the data with known SPD starts was 0.0375 (95% confidence limits 0.0224–0.0575), which is similar to the equivalent estimate derived from all the SPD data, regardless of series truncation, $\theta = 0.0308$ (95% confidence limits 0.0220–0.0415) (Fig 2B). Because of this similarity, and the greater precision of the estimate based upon all the data, we decided to use the latter in further analyses. Based upon this value of $\theta$, the arithmetic mean duration of SPDs expected if surveillance had been of indefinite duration, was 31.9 years (95% confidence limits 23.6–44.9) and 10% of SPDs would be expected to have durations exceeding 73.5 years (95% confidence limits 54.3–103.4).

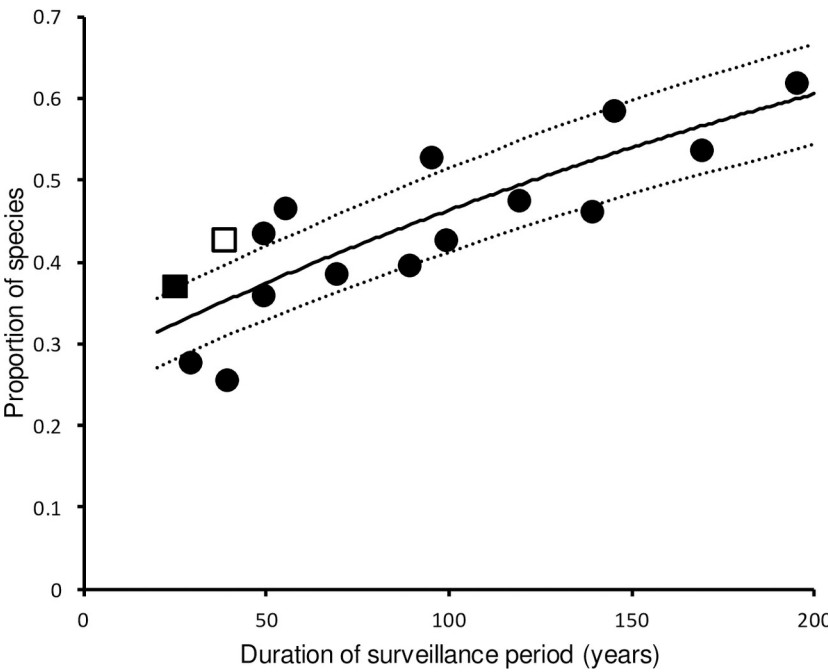

**Fig 1. Proportion of UK breeding bird species with at least one assessment period in which there was a population decline in relation to the duration of the composite surveillance period over which population status was assessed.** Filled symbols represent results based upon the population trend scores from Gibbons, Avery & Brown [21] (GAB). The filled square is for the most recent of the GAB periods (1970–1995), which overlaps with the period of our analysis of recent population trends (1970–2013). The proportion of species with an SPD, according to our definition, in 1970–2013 is shown by the open square and is plotted at the mean duration of surveillance of the 187 monitored populations for this period. All negative GAB trend scores were classified as declines.

## SPD magnitudes and annual rates of population decline

SPD magnitudes tended to increase with their duration (Fig 3). Because mean annual rates of population decline $v$ calculated from these magnitudes and durations showed no tendency for annual rate to vary with duration (see Methods), we considered it reasonable to model magnitudes as resulting from annual decline rates that vary among species, but not with SPD duration. A fitted log-normal distribution of $v$, obtained from the mean (-3.287) and standard deviation (0.710) of $\log_e$ transformed $v$ values, matched the observed cumulative distribution of values well (Fig 4: Kolmogorov-Smirnov $D = 0.080$, $P > 0.20$). The mean is equivalent to a geometric mean annual rate of population decline of 0.0374 (95% confidence limits, 0.0316–0.0430).

## Exceedance distribution of SPD magnitudes beginning in a surveillance period of 100 years

The exceedance distribution of SPD magnitudes expected in a simulated surveillance period of 100 years, derived from our models of the prevalence, duration and magnitude of SPDs is shown in Fig 5. This probability distribution can then be used to calculate the SPD magnitude that would be expected to be exceeded by a specified proportion of species during a 100-year surveillance period. For example, we estimated that 10% of species would begin an SPD of magnitude 15.8 or more in a surveillance period of 100 years (95% confidence limits of the magnitude, 8.1–42.9).

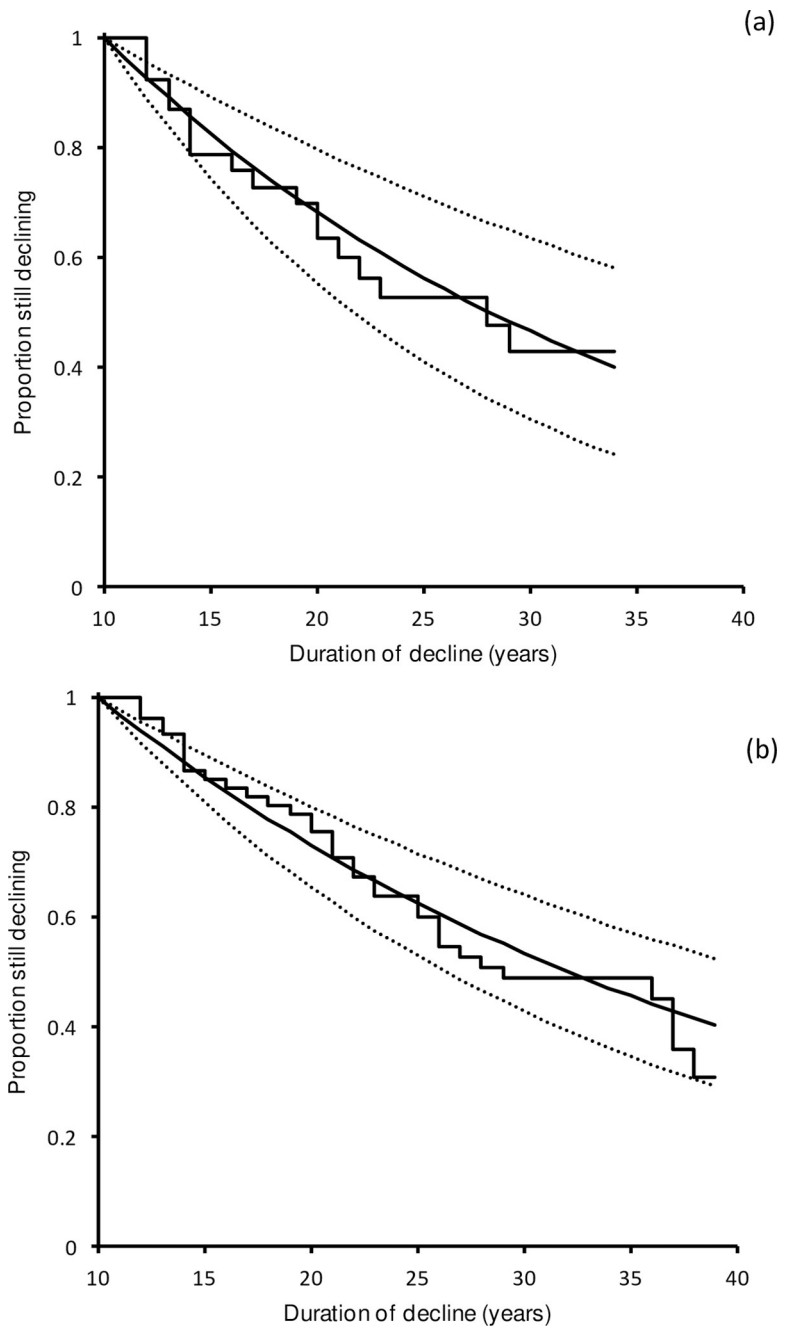

**Fig 2. Kaplan-Meier diagrams showing the proportion of SPDs of UK birds in the period 1970–2013 that were still in progress, in relation to the number of years elapsed since they began or were first detected.** The upper panel (a) shows the proportion of declines still in progress for the 43 declines with a well-defined start date. The solid curve shows the fitted exponential maximum-likelihood model. The dotted curves are 95% bootstrap confidence limits. The lower panel (b) shows the proportion of declines still in progress for all 82 declines in relation to time elapsed since the decline started or the beginning of the time series, including data for SPDs for which the start date was not well-defined.

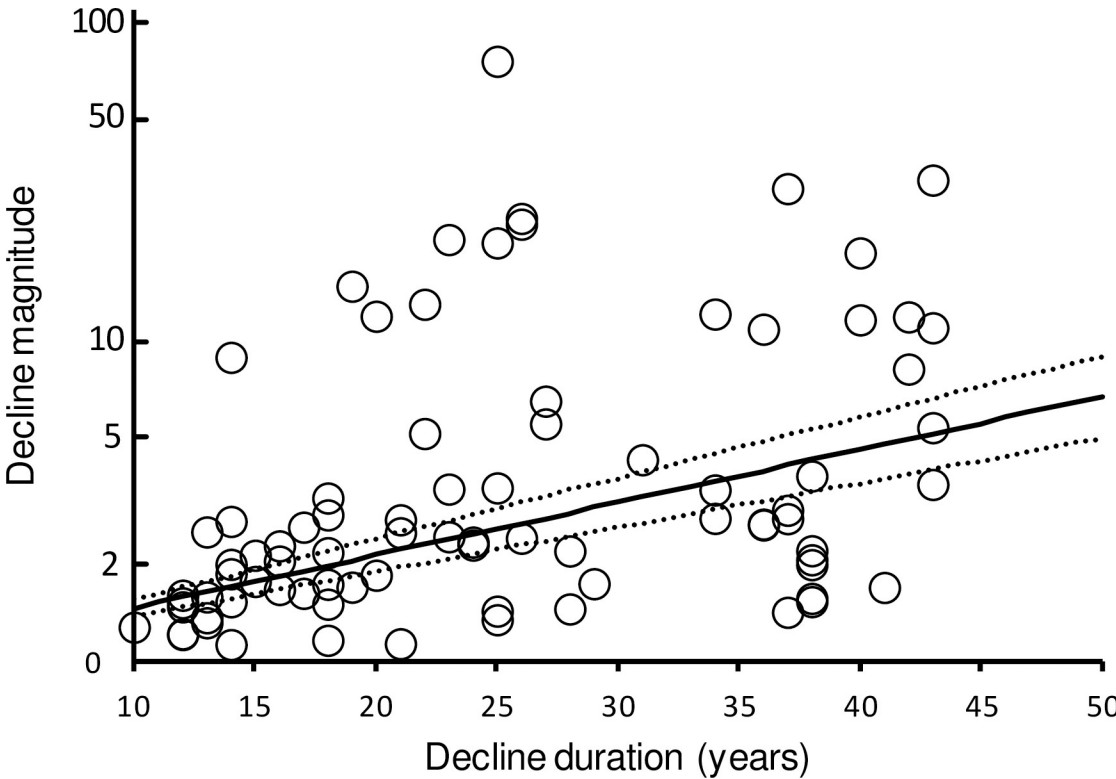

**Fig 3. Magnitudes of SPDs in relation to their duration for 82 SPDs of UK breeding bird populations during the period 1970–2013.** The vertical scale is logarithmic. The solid line represents expected values of geometric mean decline magnitude for a given decline duration, based upon the estimated mean of log-transformed annual rate of population decline averaged over all populations. The dotted lines represent lower and upper 95% bootstrap confidence limits for the modelled geometric mean decline magnitude.

## Discussion

### Prevalence, duration and magnitude of sustained population declines of UK breeding birds

Annual rates of population decline in our study are broadly comparable with those reported for declining populations from other large compilations of data on population growth rates of birds [25]. However, the true magnitude of population declines, from beginning to end, has been assessed less often because the duration of consistent quantitative monitoring is usually too short to allow the beginning and end of declines to be defined for enough populations. This can lead to underestimation of decline duration and magnitude. Our study addressed this problem by taking truncation of quantitative population time series into account in the analyses, and by calibrating quantitative analyses based upon a relatively short (43-year) time series against a qualitative assessment of population trends of UK breeding birds over almost 200 years. Our results indicate that sustained population declines (SPDs) of 10 years or more have occurred frequently in UK bird species during the last two centuries and that a substantial proportion of species is liable to declines of large magnitude. We estimated that 10% of species would be expected to begin a decline of at least sixteen-fold (i.e. a decline to not more than 6% of the initial value) during a period of 100 years. If we take the lower 95% confidence bound of the multiplier, the decline magnitude expected for 10% of species is eight-fold (a decline to not more than 12% of the initial value). If we adopt a precautionary approach and take the upper

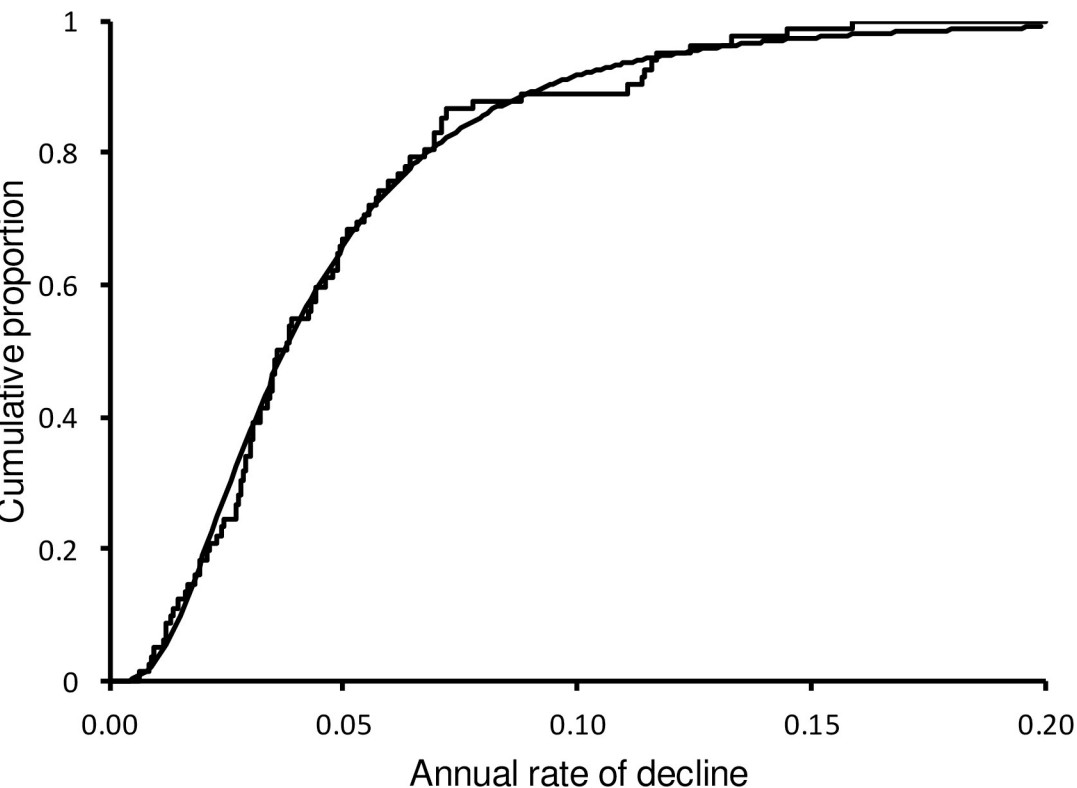

**Fig 4. Cumulative distribution (stepped line) of the mean annual rate of population decline observed for 82 SPDs recorded for UK breeding bird populations during the period 1970–2013.** The curve represents the log-normal distribution fitted by calculating least-squares estimates of the mean and standard deviation of $\log_e$-transformed annual rates of population decline.

95% confidence bound, the decline magnitude expected for 10% of species is 43-fold (a decline to about 2% of the initial value).

## Implications of sustained population declines for conservation management and the setting of a threshold for favourable population size

In birds and other taxa, population declines tend to be followed by resumed further declines or lack of recovery [26]. This implies that a high prevalence of large declines may risk moving many populations from being large and insensitive to stochastic fluctuations to being small enough to be at risk of extinction because of demographic and environmental stochasticity, even if they become stable: the small-population paradigm of Caughley [18]. Any minimum threshold measure for population size should therefore safeguard species against plausible risks of sustained population decline that might drive population size to its small-population MVP or below, in addition to guarding against the effects of environmental and demographic stochasticity once the population is at or below the small-population MVP.

Consequently, although MVPs based on the small-population paradigm and short runs of data are often all that is available for assessing extinction risk [27], our findings of a high preva-lence of large magnitude population declines suggest that such MVPs under-estimate extinc-tion risk because even a large and apparently stable population runs an appreciable risk of being subject to unforeseen long-term decline. The decline to global extinction of the passen-ger pigeon *Ectopistes migratorius* is a salutary example, as noted by Flather *et al.* [28]. This impact of risk of sustained population decline is likely to explain the finding by Reed *et al.* [27]

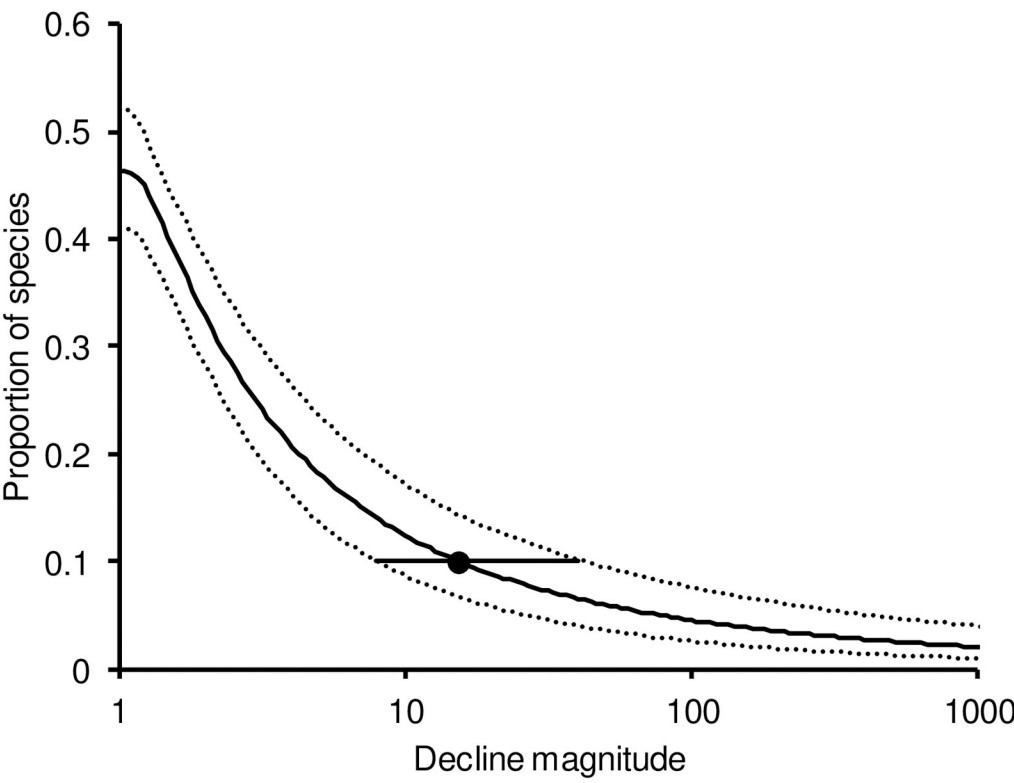

**Fig 5. Modelled exceedance (negative cumulative) distribution (solid curve) representing the proportion of all species, including those that did not decline, expected to have SPDs equal to or greater than the decline magnitude shown on the horizontal axis, with a surveillance period of 100 years.** The dotted lines show 95% bootstrap confidence limits. The filled circle represents the estimated decline magnitude exceeded by one-tenth of all species. The horizontal line is the 95% bootstrap confidence interval of this estimate.

that MVP estimates based upon studies of long duration tended to be much larger than those based upon short studies. We therefore propose that, to calculate minimum population thresholds for any currently stable population, the small-population paradigm MVP believed to be most appropriate for the focal population, should be multiplied by a factor intended to account for the risk of a sustained decline which might occur over a longer period, derived as we have illustrated for UK breeding birds. We term this factor, the *MVP Multiplier*. For example, if a stable- MVP for a population was calculated to be 1,000 adult individuals (at 90% probability of persistence for 100 years), we propose multiplying that MVP by a factor that would achieve the same probability over the same period of ensuring that the population would not be depleted to below its MVP. In the case of UK breeding birds, that factor would be 15.8, and a population of 15,800 adults could be regarded as a minimum threshold. These criteria are quantitative, but it should be recognised that the level of risk selected is arbitrary. For example, if decision-makers were unwilling to accept a risk as high as 10% that the focal population would suffer an SPD that would deplete it to below the MVP, then the multiplier value chosen would need to be greater. All existing classifications of conservation status also have arbitrary criteria based upon a human value judgement about what risks of harm to populations of wild species are acceptable. An additional consideration is the wide confidence interval for our estimate of the multiplier. A precautionary argument could be made that the upper bound of its 95% confidence limit (42.9) should be used. Future improvements in data and estimation methods might then allow the multiplier value in use to be reduced.

Which value of the multiplier should be used for a particular species or group of species requires further research. The future risk of a population undergoing a sustained decline and the decline magnitude may prove to be predictable to some extent, based upon species-specific life-history or ecological variables or projections of anthropogenic pressures such as habitat loss, pollution and climate change. To date, however, a high proportion of large sustained population declines in recent history have not been predicted, based upon either formal analyses or expert judgement. For example, no-one foresaw the recent thousand-fold decline in the global population of the white-rumped vulture *Gyps bengalensis* caused by the introduction of a veterinary non-steroidal anti-inflammatory drug [29] or the near extirpation of many populations of fish-eating birds and birds of prey in the late 20th Century caused by organochlorine pesticides [30]. For the time being, therefore, we suggest that an average value of the multiplier should be used, derived from empirical data on the prevalence and magnitude of documented declines based upon data for large groups of species. We envisage that the most feasible refinement of our method to account for reliable predictors of the future prevalence and magnitude of sustained population declines would be to incorporate predictions based upon models of species' distribution in relation to bioclimate variables. Such models have been found to provide some predictive power when observed bird population trends in Europe and the USA were compared with retrodicted trends based upon bioclimate models and observed climatic changes since the 1980s [31]. However, models of this kind have not yet been adapted to predict the magnitude of sustained population declines from predicted future climatic changes. Our suggested approach is intended to supplement rather than replace existing methods that assess the risk of global or population extinction. The prior application of established methods for the assessment of global extinction risk using established red-listing methods is essential [32], and applications of this approach at both national and local closed population levels are also practical and valuable [33], and species listed as Vulnerable, Endangered or Critically Endangered by the red-listing process should be regarded as in unfavourable conservation status without recourse to MVP Multiplier calculations. In addition, we suggest that populations of any species that are still undergoing sustained population decline at the end of a surveillance period should also be regarded as in unfavourable conservation status, regardless of their Red List status. This is because, as our results demonstrate, the eventual magnitude of any individual ongoing decline is difficult to predict and may be large.

Using these approaches sequentially, only stable and increasing populations that do not qualify as threatened under IUCN criteria and are also larger than the threshold indicated by the MVP Multiplier method would be regarded as exceeding a minimum threshold population size and, depending on other metrics, such as species range and habitat, could be classed as at Favourable Conservation Status (FCS) as defined by the European Habitats Directive 92/43/EEC. This assessment is only based upon extinction risk. Although adopting this definition based upon reduction of extinction risk would be in accord with the European Habitats Directive's requirement that a species' population with FCS should be able to "maintain itself" on a "long-term basis", fulfilling this criterion may not be an adequate condition for assigning FCS on its own. Long-term conservation success for a species is likely to require resilience to future climate change and other environmental changes which may be more frequent and have larger impacts than those which caused the past population declines that we have analysed here. Future conservation is therefore likely to require the long-term maintenance of multiple populations across the range of the species in representative ecological settings, with replicate populations in each setting, all of which should be self-sustaining, healthy, and genetically robust [34]. This logic implies that the maintenance of multiple sub-populations of a species, each of which is at a level larger than the threshold indicated by the MVP Multiplier method, would be needed before the status of the species as a whole could be regarded as favourable. However,

more work is needed to specify how many replicate conserved populations are needed and which ecological settings should be represented. Hence, we argue that the MVP Multiplier method is a valuable starting point.

## European legal interpretation of the relationship between measured criteria and FCS

FCS, as determined by a process using multiple criteria, is recommended to be a "legally binding minimum standard" [30]. Further targets for population recovery and conservation, which are independent of an FCS decision-making process and determined at national, flyway or international scales should also include multiple criteria and take account of species range and habitat in their calculation. Such a conservation target for population size is likely to be substantially greater than a minimum threshold. Correspondingly, European Commission guidance documents for the European Habitats and Birds Directives stress that FCS as a whole, must be assessed as "distance from some favourable state" rather than distance from extinction [35, 5].

## Conclusions

The MVP Multiplier method we propose in this paper offers some advantages as a method to make an extinction risk-based, minimum threshold criterion based on population size, more likely to be protective in the long term, than an MVP assessment based upon the small-population paradigm alone. We view our suggestion as providing a tool to enable a transparent and robust extinction risk based decision amongst other necessary decisions to determine FCS, as a legally binding minimum standard for species conservation. Our proposal is a response to Caughley's [18] criticism that the declining-population paradigm and small-population paradigm are rarely brought together in effective and useful ways to solve conservation problems. However, implementation of the MVP Multiplier approach requires the calculation of an appropriate value for a stable-population paradigm MVP, which remains challenging [28]. In addition, we agree with Redford *et al.* [34] that the conservation of multiple sub-populations of a species in each of several representative ecological settings is valuable. This calls for conservation assessments of widely distributed species to be undertaken at larger spatial scales than is customary at present. Finally, we recognise that there are characteristics of the status of populations relevant to Favourable Conservation Status that are only weakly linked to extinction risk and encourage the development of quantitative criteria that reflect them. These might include the degree to which the geographical range of a species covers its potential range, as determined by prevailing climatic conditions and habitat conditions in the absence of anthropogenic changes, such as pollution, overexploitation and habitat conversion.

## Supporting information

**S1 Table. Data on population changes of UK breeding bird species with more than 10 years of monitoring data available between 1970 and 2013.** Columns show: the total length in years of the monitoring data for each species; the first and last years of total monitoring period and the original dataset sources are given. Source codes are: BoCC - smoothed trends fitted to annual data from the British Trust for Ornithology (BTO and Joint Nature Conservation Committee (JNCC) Common Birds Census (CBC) and the BTO/JNCC/Royal Society for the Protection of Birds (RSPB) Breeding Bird Survey (BBS) (Harris et al. 2015); RBBP - data from the Rare Breeding Birds Panel (Holling 2014); SCARRABS - national population estimates from surveys of single species, which were typically undertaken at intervals of at least several years (Eaton et al. 2015); Declines of at least 10 years duration were identified, and the

first ($t_1$) and last ($t_2$) years of the declines are given. The population size or indexed population sizes $n_1$ at $t_1$ and $n_2$ at $t_2$ are given. References are as supplied in the main text.
(DOCX)

## Acknowledgments

We thank Paul Donald and Mark Eaton for useful discussions and Joseph Veech and an anonymous reviewer for constructive criticisms.

## Author Contributions

**Conceptualization:** Rhys E. Green.

**Data curation:** Gillian Gilbert.

**Formal analysis:** Rhys E. Green.

**Investigation:** Gillian Gilbert, Jeremy D. Wilson, Kate Jennings.

**Methodology:** Rhys E. Green, Jeremy D. Wilson.

**Project administration:** Gillian Gilbert.

**Supervision:** Gillian Gilbert.

**Validation:** Rhys E. Green.

**Visualization:** Rhys E. Green, Gillian Gilbert, Jeremy D. Wilson, Kate Jennings.

**Writing – original draft:** Rhys E. Green.

**Writing – review & editing:** Gillian Gilbert, Jeremy D. Wilson, Kate Jennings.

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
