## [Decision Letter · Decision Letter 0]

25 Oct 2019

PONE-D-19-27507

Implications of the prevalence and magnitude of sustained declines for determining a minimum threshold Favourable Reference Value for population size

PLOS ONE

Dear Dr. Gilbert,

Thank you for submitting your manuscript to PLOS ONE. After careful consideration, we feel that it has merit but does not fully meet PLOS ONE’s publication criteria as it currently stands. Therefore, we invite you to submit a revised version of the manuscript that addresses the points raised during the review process.

Both reviewers found the topic of the manuscript interesting and both indicated that the writing could be improved to increase clarity and focus. I agree with this assessment. Please consider all comments of both reviewers carefully when revising the manuscript.

We would appreciate receiving your revised manuscript by January 30, 2019. To enhance the reproducibility of your results, we recommend that if applicable you deposit your laboratory protocols in protocols.io, where a protocol can be assigned its own identifier (DOI) such that it can be cited independently in the future. For instructions see: http://journals.plos.org/plosone/s/submission-guidelines#loc-laboratory-protocols

We look forward to receiving your revised manuscript.

Kind regards,

Floyd W Weckerly

Academic Editor

PLOS ONE

**Journal Requirements:**

**Comments to the Author**

1. Is the manuscript technically sound, and do the data support the conclusions?

Reviewer #1: Yes

Reviewer #2: Partly

2. Has the statistical analysis been performed appropriately and rigorously? 

Reviewer #1: Yes

Reviewer #2: No

3. Have the authors made all data underlying the findings in their manuscript fully available?

Reviewer #1: Yes

Reviewer #2: Yes

4. Is the manuscript presented in an intelligible fashion and written in standard English?

Reviewer #1: Yes

Reviewer #2: No

5. Review Comments to the Author

Reviewer #1: I have carefully read PONE-D-19-27507 “Implications of the prevalence and magnitude of sustained declines for determining a minimum threshold Favourable Reference Value for population size” by Green et al. and now provide this review. Overall, the manuscript is well written, design and analysis are sound, and the goal of the study is very worthwhile. My comments mostly pertain to increasing the clarity of the writing in some places and also expanding the context to make the manuscript of interest to a greater readership. Comments are presented in order of occurrence in manuscript.

Line 30: “animal populations”, not just birds? See my later comment about the potential difficulty of applying this technique for organisms in which there is not much long-term monitoring data.

Line 35: Is this literally true? We would hope that the multiplier can reduce the risk that population gets depleted below MVP, but whether population actually falls below MVP depends on events out there in the real world, not whether or not an MVP value is increased via the multiplier.

Lines 38 – 46: These lines are referring to all the bird species collectively, is this correct? That is, the multiplier is not calculated separately for each species. Writing needs to be clearer. (Later in manuscript this is clear, but needs to be so here as well.)

Lines 40 – 41: This is a bit ambiguous. Does the decline of 16-fold refer to a population decreasing by 16X over the course of 100 years? I think it would be more direct and easier for the reader to comprehend if the authors thought about population declines as percent change per year. For instance, even a consistent 2 - 3% decline per year is very substantial if it occurs over many decades. If I did the math right, a decline of 16X over 100 years corresponds to an annual decline of 2.8%. Later in the manuscript, the authors do refer to percent decline per year, but need some mention here as well.

Except for Lines 63 - 65, Introduction is very Euro-focused. That's ok, but manuscript would have more appeal if it could also refer (briefly) to conservation legislation is USA/Canada.

Line 57: Authors should see the Partners-in-Flight Landbird Population Estimates Database and Rosenberg et al. (2016) paper. But be aware that the methodology of the Rosenberg et al. paper was criticized. Nonetheless, it is an example (North American) of attempting to estimate range-wide population sizes and set target values.

Lines 63 – 65: This is a correct statement. However, there is some conservation legislation that does specify size of a recovered population. The Marine Mammal Protection Act (USA) specifies that marine mammal populations must be maintained (managed) to be at a size that is at least half the carrying capacity. (Of course, the problem is that it is difficult to estimate K.) Also, with regard to this sentence, it might be better to say that the methods do not always identify a target population size (for recovery, healthy, favourable) rather than saying that the methods are "poorly developed".

Lines 71 – 76: Agreed, well-stated.

Lines 84 – 85: Again, this sentence seems to imply that the multiplier is not calculated separately for each species. Writing needs to be more explicit and precise. (Granted, later in the manuscript, this ambiguity is cleared up.)

Lines 95 – 97: This sentence sort of leaves the reader hanging wondering what this two-century population trend study is all about. Perhaps add one or two more sentences that give a little more information, even though the study is described a little more later in the manuscript.

Line 120 and elsewhere: The overall idea of a SPD and being able to identify one quantitatively is interesting. Have other authors written about this, and particularly used the phrase "sustained population decline"?

Lines 128-137: It might be worthwhile to have a figure that helps explain the determination (identification) of a SPD. Also, have any other authors used these criteria? They seem straightforward enough that other authors might have used same or very similar, and hence should be cited.

Lines 135 – 137: Does this mean that a given species might have more than one SPD? If so, say this explicitly. (I saw later that Line 147 answered my question, nonetheless, provide a little more info here.)

Lines 142-144: OK, good to identify these censored SPDs, but then what? Were they still used in subsequent analyses? Might need one brief follow-up sentence here. Line 155-156: OK, but again, how was this information used? What did you do with these truncated SPDs? Line 157: Again, this still begs the question of why truncated SPDs were identified as such.

Lines 168-170: More precisely, I think this is an exponential DECAY function.

Line 172: More precisely, is theta the annual PROBABILITY of an SPD ending? Be consistent with Line 165.

Lines 179-181: Not sure what the authors mean by observed and expected numbers of declines. Is the expected number based on a given estimated value of theta? and hence the chi-square test is a way of assessing if the estimated value of theta is "accurate", as in a non-significant chi-square statistic? More explanation may be needed.

Lines 220-222: Doesn't the prevalence of SPDs also need to take into account the magnitude of an SPD, or is this "built into" the definition of an SPD? That is, an SPD can have any m value greater than zero, correct? (Nonetheless, identifying an SPD in count data from an annual survey probably is made more likely as the severity of the decline in the real population becomes more pronounced, and as such the observed prevalence of SPDs would depend on their estimated magnitude because this itself depends on the real magnitude.)

Line 243: Proportion of composite periods? or proportion of species (showing an SPD) during the composite period?

Lines 242 - 252: I'm not sure that the 15 surveillance periods of the GAB data are truly independent of one another with regard to the response variable, proportion of species with a decline. For example, when two periods of length a and b are combined to give a period with length c, doesn't the proportion of species with a decline in period c somehow depend on the proportions in a and b? I don't know for sure if there's an issue here with non-independence, but if there is, then it would call into question the legitimacy of the regression (Fig. 1).

Lines 262 - 263: Over what range of magnitudes? Line 268: A decline of what magnitude?

Line 271: Species as the unit for bootstrapping? or more precisely, was it the observed SPDs that were resampled (bootstrapped)?

Lines 325 - 328: These results hold regardless of magnitude of the SPD? That is, the values of 31.9 years, 10%, and 73.5% were determined over a wide range of magnitudes?

Lines 389 - 391: I'm not familiar with the reference that the authors give. In addition, they should see some of the research using the North American Breeding Bird Survey data, these are publications by John Sauer and others.

Lines 394-395: Be more precise about what is meant by "negative bias". I think the authors are suggesting that population declines (duration and hence overall magnitude) might be UNDERESTIMATED when the beginning and end of the trends cannot be identified in the monitoring data.

Lines 395-396: A recurring theme in the manuscript is that time-series data are often left and right-truncated and that this must be taken into account when trying to quantify a trend. To me, it seems like this must be a relatively common issue in the use of time-series data, are there general (and statistical) references to cite? Also, how have other authors dealt with this issue, either in bird data or perhaps population data (or any time series data) for other animal taxa?

Lines 402 - 407: As I previously commented, to me it makes more sense and is more intuitive to cast the trend as percent population change per year rather than (or in addition to) an X-fold difference between beginning and ending population size. A decline of 2 - 3% is serious when it is continual (every year) and over a long period of time (decades to century). This is what this study is really getting at. 8-fold for 100 years = -2.1% per year, 43-fold = -3.8%.

Lines 422 - 426: Excellent point.

Lines 430 - 458: I generally agree with everything that is stated here. However, I think the authors need to (1) not rely so much on the 15.8 multiplier that would be needed to offset 10% of birds expected to have an SPD of this magnitude within 100 years. It is ok to present this info but perhaps briefly present other scenarios (e.g., X% of birds expected to have an SPD of magnitude 5 or greater over a time period of 50 years). Perhaps a table of values for different multiplier scenarios could be useful. (2) Putting this analytical technique into practice for other taxa could be very difficult. Other than birds (and perhaps some insects) there is not any really long-term data that could be used to derive the multiplier. Authors should acknowledge this limitation more thoroughly and perhaps include an appeal for ecologists to start collecting such long-term data on other taxa.

Lines 479-482: Another good point.

Lines 493-519: This section is also very Euro-focused. That's ok but it does limit the appeal of the study somewhat. Perhaps the authors could link to conservation legislation in USA/Canada a little bit.

Lines 504-519: Sentences here seem a little off the subject or out of place. None of this content derives directly from the results of this study (calculation and values of the multiplier). At the least, these sentences should be condensed or incorporated into Introduction.

Overall the conclusion section is good, however be careful with statements such as Lines 529-530 "one transparent and robust population size decision". Is the "one" referring to using just one value of a multiplier? This would be a bad idea when defining the size of a recovered population. Rather, it is better to use a range of multipliers (e.g., 95% CIs) for adjusting the MVP, just as no one would present just one MVP size.

I enjoyed reading this manuscript.

Joe Veech

Reviewer #2: I find this manuscript very difficult to review. It contains an extensive analysis of sustained population declines in UK birds, and applies the results of that analysis to recommend adjustments to minimum population size (FRV-P) values that are part of the FCS assessments of the EU. I have concerns about the overall integration of the various components of the manuscript.

A few general comments:

1. It is really difficult to decipher the alphabet soup of acronyms in the manuscript.

2. These is a fundamental fuzziness in much of the descriptions of the conservation assessment information (such as FRV, FRV-P, etc) that complicates this discussion. Is FRV-P a threshold category for conservation action or a population status? That is never mentioned.

3. The primary analysis of the manuscript is poorly motivated. After a discussion of EU conservation categorizations, the manuscript switches gear to provide an extensive discussion of modeling durations of population declines. Then, it only returns to the original motivation (the conservation assessments) at the end, applying the very general results to attempt to inform very specific questions. These pieces need better integration.

It seems to me that there is a disconnection between the abstract (that is strongly focused on the conservation application) and the bulk of the manuscript (that is primary about the analysis of sustained population declines). I find it difficult to believe that the high-level result extracted from the bird population analysis and applied to conservation question (i.e., how much to adjust FRV-P) really has credibility, as it is produced at such a high level of abstraction from any species.

I question the rather grandiose assessment that this approach combines Caughley's rare population and declining population paradigms. I would that would best be done in a model that contains temporal stochasticity in demographic parameters.

4. I find it difficult to pass judgement on the quality of the primary analysis, of sustained population declines. It is an interesting result, but it seems to me that there are many assumptions that need to be stated. There would be huge heterogeneity among species in this probability, but that is not considered here, and the way significant periods of decline is defined and estimated makes no accommodation for the imprecision inherent in estimation of declines.

Specific comments:

Abstract: I find this to be quite confusing, as it pummels the readers with poorly described jargon. It could use some strong editing to provide a clear statement of the issue and how this analysis improves it. As I note elsewhere, it seems to me that the focus of the paper is the analysis of sustained declines, but the focus of the abstract is how the overall result ties into 1 component of FCS.

L 28. I had not heard of the term Favorable Conservation Status, and this first sentence just does'nt make sense to me. This should be reworded to at last describe (even vaguely!) what FCS is, and perhaps how FRV-P (l. 30) relates to it!

l. 55. Quantitative methods are always needed! I think you want to say that efficient methods are needed.

l. 59. You still have not defined this concept of "Favorable Conservation Status!" What does it mean? L 66-69 indicate that this concept relates to criteria we usually asspciate with risks of extinction, or species characteristics that provide a rationale for conservation activity. Please state this.

l. 69 and following. You choose to focus on just one element of FCS. It seems to me that your introduction up to this point is misleading, as you are not discussing FCS comprehensively, but are just refining one element of it.

l. 71 and following. Pleae clarify the motivation for defining FRV-P. There seems to be a lot of semantics on this paragraph. You call this "numerical measures of population size." Is this the current population size (i.e, the population status), or is this a minimum viable population size? I find this confusing. Defining population status is common, and identifying species with small populations is often considered a risk factor for conservation assessments (i.e., Rabinowitz-style criteria), but these status values are much different than MPVs which relate specifically to a demographic modeling attribute. Statements in this paragraph suggest that you wish to define FRV-P as a modified MVP, but is that the actual intent of the FRV-P criterion?

I believe that what you are saying here is the FRV-P is designed to be a MVP, and that your analysis provides an ad-hoc scaling of MVP to accommpdate stochastic effects not included in the original MVP formulation. If so, why not just define a more realistic MVP that includes such stockasticity?

l. 90-93. This seems to me to be a odd comment. How do we know that several decades is short relative to the duration of some declines? It would be better to say that perhaps declines defined over short intervals are not predictive of long term declines and thus do not inform extinction probabilities (as Thogmartin and Stanton have done for North American birds).

l. 121-123. This criterion defines a period of decline and recovery. the later criteria (up to line 135) are rules for finding the smallest n in the interval. Why not just say that?

l. 138. You never state it explicitly, but presumably the duration of the SPD is from tstart to tlow.

SPDs must also be at least 10 years in duration. This seems very arbitrary, and likely to have a strong influence on the results.

l. 165 onward. All analyses after this point make identical distribution assumptions among the SBDs. Is that valid?

l. 165-170. This seems to me to be an oversimplification. If so, why only consider d >9? Are there issues associated with not having d <10 in this analysis?

6. PLOS authors have the option to publish the peer review history of their article (what does this mean?). If published, this will include your full peer review and any attached files.

Reviewer #1: Yes: Joseph Veech

Reviewer #2: No

---

## [Author Response · Author response to Decision Letter 0]

17 Jan 2020

Dear Editor, the following is a copy of our Response to Reviewers' comments, I hope that is what is required in this section.

Rebuttal letter for manuscript: PONE-D-19-27507

Implications of the prevalence and magnitude of sustained declines for determining a minimum threshold Favourable Reference Value for population size.

AUTHORS’ INTRODUCTION TO THE REBUTTAL

We thank the reviewers for their comments. In the following list, we have numbered all the comments which require a response (labelled COMMENT) and then responded to each one (labelled RESPONSE).

COMMENT 1.1

Reviewer #1: I have carefully read PONE-D-19-27507 “Implications of the prevalence and magnitude of sustained declines for determining a minimum threshold Favourable Reference Value for population size” by Green et al. and now provide this review. Overall, the manuscript is well written, design and analysis are sound, and the goal of the study is very worthwhile. My comments mostly pertain to increasing the clarity of the writing in some places and also expanding the context to make the manuscript of interest to a greater readership. Comments are presented in order of occurrence in manuscript.

RESPONSE: We appreciate this comment very much and are most grateful for the reviewer’s careful reading of our MS. The constructive and perceptive criticisms have been most valuable in making revisions.

COMMENT 1.2

Line 30: “animal populations”, not just birds? See my later comment about the potential difficulty of applying this technique for organisms in which there is not much long-term monitoring data.

RESPONSE: We accept that data availability limitations will affect the taxa to which our suggested approach will be applicable. However, we think that there is nothing about our approach that is specific to birds, given that time-series data on the size of populations or indices of population size can be obtained. We think that such data of this kind exist for more than just birds, as is illustrated by the taxonomic coverage of, for example, the Living Planet Index. We have revised the text in the Introduction to say this. 

COMMENT 1.3

Line 35: Is this literally true? We would hope that the multiplier can reduce the risk that population gets depleted below MVP, but whether population actually falls below MVP depends on events out there in the real world, not whether or not an MVP value is increased via the multiplier.

RESPONSE: Thank you for pointing out this sloppy drafting. We intended to say that the approach ALLOWS FOR the risk, not that it reduces it. We have rewritten that section to correct this.

COMMENT 1.4

Lines 38 – 46: These lines are referring to all the bird species collectively, is this correct? That is, the multiplier is not calculated separately for each species. Writing needs to be clearer. (Later in manuscript this is clear, but needs to be so here as well.)

RESPONSE: Thank you for pointing this out. You are correct and we have clarified the text here.

COMMENT 1.5

Lines 40 – 41: This is a bit ambiguous. Does the decline of 16-fold refer to a population decreasing by 16X over the course of 100 years? I think it would be more direct and easier for the reader to comprehend if the authors thought about population declines as percent change per year. For instance, even a consistent 2 - 3% decline per year is very substantial if it occurs over many decades. If I did the math right, a decline of 16X over 100 years corresponds to an annual decline of 2.8%. Later in the manuscript, the authors do refer to percent decline per year, but need some mention here as well.

RESPONSE: We wrote “Over a surveillance period of 100 years, we estimated that there was a 10% risk across species, that a sustained population decline of at least sixteen-fold would begin.” Hence, we are not concluding that a population would decline by that amount within that period. We believe this to be clear and that also including the rate of annual decline may be more confusing. Our logic for focussing of decline magnitudes rather than rates is explained later in the paper.

COMMENT 1.6

Except for Lines 63 - 65, Introduction is very Euro-focused. That's ok, but manuscript would have more appeal if it could also refer (briefly) to conservation legislation is USA/Canada.

RESPONSE: We agree. Our paper was stimulated by concern among people interested in the European Union’s environmental regulations and standards. We have revised the text throughout to broaden the focus. For example, we have included a reference to Partners in Flight population targets (see below). 

COMMENT 1.7

Line 57: Authors should see the Partners-in-Flight Landbird Population Estimates Database and Rosenberg et al. (2016) paper. But be aware that the methodology of the Rosenberg et al. paper was criticized. Nonetheless, it is an example (North American) of attempting to estimate range-wide population sizes and set target values.

RESPONSE: Thank-you for this suggestion which has been included in the text

COMMENT 1.8

Lines 63 – 65: This is a correct statement. However, there is some conservation legislation that does specify size of a recovered population. The Marine Mammal Protection Act (USA) specifies that marine mammal populations must be maintained (managed) to be at a size that is at least half the carrying capacity. (Of course, the problem is that it is difficult to estimate K.) Also, with regard to this sentence, it might be better to say that the methods do not always identify a target population size (for recovery, healthy, favourable) rather than saying that the methods are "poorly developed".

RESPONSE: We agree and have revised the Introduction to be clearer on this, although we have not directly included the Marine Mammal protection Act example.

COMMENT 1.9

Lines 71 – 76: Agreed, well-stated.

RESPONSE: We appreciate this positive comment. We have retained text on this key principle concerning the Caughley small-population paradigm MVP, though we have restructured the section somewhat to accommodate other issues.

COMMENT 1.10

Lines 84 – 85: Again, this sentence seems to imply that the multiplier is not calculated separately for each species. Writing needs to be more explicit and precise. (Granted, later in the manuscript, this ambiguity is cleared up.)

RESPONSE: We have changed the wording to clarify that calculation for each species is necessary.

COMMENT 1.11

Lines 95 – 97: This sentence sort of leaves the reader hanging wondering what this two-century population trend study is all about. Perhaps add one or two more sentences that give a little more information, even though the study is described a little more later in the manuscript.

RESPONSE: We have improved this section, but aimed not to repeat what (as the reviewer acknowledges) we say later.

COMMENT 1.12

Line 120 and elsewhere: The overall idea of a SPD and being able to identify one quantitatively is interesting. Have other authors written about this, and particularly used the phrase "sustained population decline"?

RESPONSE: We think that this is a novel term , though we think that the concept is logical, simple and useful.

COMMENT 1.13

Lines 128-137: It might be worthwhile to have a figure that helps explain the determination (identification) of a SPD. Also, have any other authors used these criteria? They seem straightforward enough that other authors might have used same or very similar, and hence should be cited.

RESPONSE: We considered this but have not come up with a clear enough figure. A schematic diagram can be more ambiguous than the description in the written text. The algorithmic determination of an SPD in this paper is original and was not taken from another published text. Although it is complicated, we think that having full details of the definition as text is the best way to present it.

COMMENT 1.14

Lines 135 – 137: Does this mean that a given species might have more than one SPD? If so, say this explicitly. (I saw later that Line 147 answered my question, nonetheless, provide a little more info here.)

RESPONSE: Yes, we have clarified this.

COMMENT 1.15

Lines 142-144: OK, good to identify these censored SPDs, but then what? Were they still used in subsequent analyses? Might need one brief follow-up sentence here. Line 155-156: OK, but again, how was this information used? What did you do with these truncated SPDs? Line 157: Again, this still begs the question of why truncated SPDs were identified as such.

RESPONSE: We have checked the text on truncation and believe that the logic is clear. Truncated SPDs cannot be used uncritically to determine duration for obvious reasons. However, before truncation has occurred the data can be used to estimate the annual probability that an SPD which is in progress comes to an end. This is a form of censored survival analysis, where “survival” in this case is the persistence from one year to the next of an ongoing decline. We have added text to make this point. All SPDs contributed some data, including values for the mean annual rate of decline, which was obtainable for all of them.

COMMENT 1.16

Lines 168-170: More precisely, I think this is an exponential DECAY function.

RESPONSE: Indeed. This is an exponential decay function. We have now added those helpful words to the sentence defining the function.

COMMENT 1.17

Line 172: More precisely, is theta the annual PROBABILITY of an SPD ending? Be consistent with Line 165.

RESPONSE: Yes, that is a clearer way of putting it, so we’ve used it. Thank you.

COMMENT 1.18

Lines 179-181: Not sure what the authors mean by observed and expected numbers of declines. Is the expected number based on a given estimated value of theta? and hence the chi-square test is a way of assessing if the estimated value of theta is "accurate", as in a non-significant chi-square statistic? More explanation may be needed.

RESPONSE: The reviewer is correct. The purpose of the chi-squared test is to check whether assuming the exponential decay function is a reasonable way to model the rate at which SPDs come to an end (by populations stopping declining and becoming stable). We have added text to explain that better.

COMMENT 1.19

Lines 220-222: Doesn't the prevalence of SPDs also need to take into account the magnitude of an SPD, or is this "built into" the definition of an SPD? That is, an SPD can have any m value greater than zero, correct? (Nonetheless, identifying an SPD in count data from an annual survey probably is made more likely as the severity of the decline in the real population becomes more pronounced, and as such the observed prevalence of SPDs would depend on their estimated magnitude because this itself depends on the real magnitude.)

RESPONSE: “Prevalence” is defined as whether an SPD occurs or not (see line 220-221) and not how big or rapid it is. We show elsewhere (lines 192-196) that annual RATE of decline in SPDs is not dependent on their duration. That allows us to estimate rates of decline and prevalences separately and then combine them later (at line 259). We have added text to say that. Thank you for pointing out the need to do this.

COMMENT 1.20

Line 243: Proportion of composite periods? or proportion of species (showing an SPD) during the composite period?

RESPONSE: We have reworded this as the reviewer suggests. This makes it clearer- thank you.

COMMENT 1.21

Lines 242 - 252: I'm not sure that the 15 surveillance periods of the GAB data are truly independent of one another with regard to the response variable, proportion of species with a decline. For example, when two periods of length a and b are combined to give a period with length c, doesn't the proportion of species with a decline in period c somehow depend on the proportions in a and b? I don't know for sure if there's an issue here with non-independence, but if there is, then it would call into question the legitimacy of the regression (Fig. 1).

RESPONSE: The 15 surveillance periods certainly are not truly independent. This is why we did not perform a significance test on the regression. Our intention here was not to perform a test of the (highly implausible a priori) null hypothesis that decline prevalence is unrelated to surveillance period duration. Rather, it was to obtain an empirical relationship between prevalence and duration. The regression is legitimate for that purpose. We have added some text to the results to say this.

COMMENT 1.22

Lines 262 - 263: Over what range of magnitudes? Line 268: A decline of what magnitude?

RESPONSE: Our analysis covers the whole range of magnitudes. We have added text to this effect.

COMMENT 1.23

Line 271: Species as the unit for bootstrapping? or more precisely, was it the observed SPDs that were resampled (bootstrapped)?

RESPONSE: it was species (as stated), but in almost all cases there was just one SPD per species, so it is nearly the same thing.

COMMENT 1.24

Lines 325 - 328: These results hold regardless of magnitude of the SPD? That is, the values of 31.9 years, 10%, and 73.5% were determined over a wide range of magnitudes?

RESPONSE: This statement refers to SPDs of all magnitudes. Decline magnitude is taken into account in a later steps in our procedure (decribed at lines 339 and 364). We show elsewhere that the rate of decline was not related to SPD duration (see lines 192-196).

COMMENT 1.25

Lines 389 - 391: I'm not familiar with the reference that the authors give. In addition, they should see some of the research using the North American Breeding Bird Survey data, these are publications by John Sauer and others.

RESPONSE: This is a useful point. We have referenced a recent Partners in Flight report which uses a method based on decline rates and numerical population targets.

COMMENT 1.26

Lines 394-395: Be more precise about what is meant by "negative bias". I think the authors are suggesting that population declines (duration and hence overall magnitude) might be UNDERESTIMATED when the beginning and end of the trends cannot be identified in the monitoring data.

RESPONSE: We have used the reviewer’s wording here to make the point clear and jargon-free. Our new text, “This can lead to underestimation of decline duration and magnitude” says it unambiguously. 

COMMENT 1.27

Lines 395-396: A recurring theme in the manuscript is that time-series data are often left and right-truncated and that this must be taken into account when trying to quantify a trend. To me, it seems like this must be a relatively common issue in the use of time-series data, are there general (and statistical) references to cite? Also, how have other authors dealt with this issue, either in bird data or perhaps population data (or any time series data) for other animal taxa?

RESPONSE: This is a good point. We have added text drawing attention to the analogy between our analyses of SPD duration and censored analyses of survivorship. The reference to Kalbfleisch (1979) gives methods.

COMMENT 1.28

Lines 402 - 407: As I previously commented, to me it makes more sense and is more intuitive to cast the trend as percent population change per year rather than (or in addition to) an X-fold difference between beginning and ending population size. A decline of 2 - 3% is serious when it is continual (every year) and over a long period of time (decades to century). This is what this study is really getting at. 8-fold for 100 years = -2.1% per year, 43-fold = -3.8%.

RESPONSE: We agree with the reviewer that it is neither the duration nor the rate of the decline that is important from the point of view of conservation, but how rate and duration combine to produce magnitude (change from beginning to end of and SPD). So we disagree with the conclusion the reviewer draws here whilst supporting the foregoing logic!

COMMENT 1.29

Lines 422 - 426: Excellent point.

RESPONSE: We appreciate the reviewer’s support here.

COMMENT 1.30

Lines 430 - 458: I generally agree with everything that is stated here. However, I think the authors need to (1) not rely so much on the 15.8 multiplier that would be needed to offset 10% of birds expected to have an SPD of this magnitude within 100 years. It is ok to present this info but perhaps briefly present other scenarios (e.g., X% of birds expected to have an SPD of magnitude 5 or greater over a time period of 50 years). Perhaps a table of values for different multiplier scenarios could be useful. (2) Putting this analytical technique into practice for other taxa could be very difficult. Other than birds (and perhaps some insects) there is not any really long-term data that could be used to derive the multiplier. Authors should acknowledge this limitation more thoroughly and perhaps include an appeal for ecologists to start collecting such long-term data on other taxa.

RESPONSE: We did not intend to rely on the 15.8 multiplier. We point out (lines 440-443) that the selection of this value of the multiplier depends upon an arbitrary decision about how much risk is acceptable- in this case we take it that a 90% chance of the population not being depleted to below the small-population paradigm MVP is acceptable. We have added some text at line 441 to emphasise this. We have added text to the Introduction about the range of taxa to which our method could be applied. We think that it is quite wide. In addition to birds and insects, mentioned by the reviewer, there is the range of vertebrate taxa covered by the Living Planet Index.

COMMENT 1.31

Lines 479-482: Another good point.

RESPONSE: We appreciate the reviewer’s support here.

COMMENT 1.32

Lines 493-519: This section is also very Euro-focused. That's ok but it does limit the appeal of the study somewhat. Perhaps the authors could link to conservation legislation in USA/Canada a little bit.

RESPONSE: We agree and have widened the focus of this section.

COMMENT 1.33

Lines 504-519: Sentences here seem a little off the subject or out of place. None of this content derives directly from the results of this study (calculation and values of the multiplier). At the least, these sentences should be condensed or incorporated into Introduction.

RESPONSE: We agree and have reduced and rewritten this section and put some of the content into the Introduction where the logic is clearer.

COMMENT 1.34

Overall the conclusion section is good, however be careful with statements such as Lines 529-530 "one transparent and robust population size decision". Is the "one" referring to using just one value of a multiplier? This would be a bad idea when defining the size of a recovered population. Rather, it is better to use a range of multipliers (e.g., 95% CIs) for adjusting the MVP, just as no one would present just one MVP size.

RESPONSE: This is a good point we have changed the ‘one’ to ‘a’.

COMMENT 1.35

I enjoyed reading this manuscript.

RESPONSE: We are most grateful for the reviewer’s careful scrutiny and constructive suggestions.

Joe Veech

REVIEWER #2: I find this manuscript very difficult to review. It contains an extensive analysis of sustained population declines in UK birds, and applies the results of that analysis to recommend adjustments to minimum population size (FRV-P) values that are part of the FCS assessments of the EU. I have concerns about the overall integration of the various components of the manuscript.

A few general comments:

COMMENT 2.1

It is really difficult to decipher the alphabet soup of acronyms in the manuscript.

RESPONSE: We have removed much of this from the manuscript to make it more accessible and generally applicable beyond the issue of Favourable Conservation Status as defined legally in Europe. 

COMMENT 2.2

These is a fundamental fuzziness in much of the descriptions of the conservation assessment information (such as FRV, FRV-P, etc) that complicates this discussion. Is FRV-P a threshold category for conservation action or a population status? That is never mentioned.

RESPONSE: We use these terms because they are familiar to those engaged with the EU process. However, we agree that this is a limited subset of readers immersed in the world of FRVs, so have generalised the wording of the conservation assessment information to widen its interest and improve clarity. We have also altered the title to remove mention of terms used in the European Union’s legal processes.

COMMENT 2.3

The primary analysis of the manuscript is poorly motivated. After a discussion of EU conservation categorizations, the manuscript switches gear to provide an extensive discussion of modeling durations of population declines. Then, it only returns to the original motivation (the conservation assessments) at the end, applying the very general results to attempt to inform very specific questions. These pieces need better integration.

RESPONSE: Our Introduction now more clearly outlines our motivation for the new method proposed. 

COMMENT 2.4

It seems to me that there is a disconnection between the abstract (that is strongly focused on the conservation application) and the bulk of the manuscript (that is primary about the analysis of sustained population declines). I find it difficult to believe that the high-level result extracted from the bird population analysis and applied to conservation question (i.e., how much to adjust FRV-P) really has credibility, as it is produced at such a high level of abstraction from any species.

RESPONSE: We have improved the abstract and the Introduction to lay a better foundation for what the paper is trying to achieve. We believe that reviewer 2, when criticising the proposed method as having a ‘high level of abstraction from any species’ is criticising the generic nature of the multiplier. We believe as stated in the manuscript, that this is a strength because it does not make assumptions as to the types of declines that may befall individual species and allows the unpredictability of real conservation problems to be common to all species. Our aim is to illustrate a generally protective multiplier to render small-population paradigm MVPs more robust, given that unpredicted long-term adverse changes befall populations. It is not intended to predict outcomes for individual species. We are highly sceptical that risks of future Sustained Population Declines can be predicted from species’ attributes, with the possible exception of effects of climatic change.

COMMENT 2.5

I question the rather grandiose assessment that this approach combines Caughley's rare population and declining population paradigms. I would that would best be done in a model that contains temporal stochasticity in demographic parameters.

RESPONSE: We disagree with the reviewer here. The reviewer’s assertion that our suggestion that our approach combines Caughley’s two paradigms is “grandiose” seems harsh and we believe is misplaced. The two paradigms are manifestly explicitly linked together in our study- SPDs are characteristic of the declining-population paradigm and the MVP values that we propose to multiply reflect the small-population paradigm. So we think it reasonable to make this simple point, especially given Caughley’s criticism that the two paradigms are too rarely brought together in solving practical conservation problems. The reviewer would prefer us to link the two in a very different way, using a PVA based upon a simulation model which uses estimates of temporal stochasticity in demographic parameters. If it was feasible, this might well be useful in assessing long-term risks to a given population. However, it would require robust estimates of all of the many parameters in the model. The data to support such models are simply not available for most species. In large part, that is because valid PVAs have to be based upon data which are not only robust, but also long-term. That combination is rarely achieved. Our different approach in this paper accepts that the future is difficult to predict and quantifies the rate of occurrence of past sustained population declines and their magnitudes, based upon empirical data. We are happy to admit that these rates and magnitudes might change over time and might not be a completely reliable guide to the future. However, we suggest that, at least in the foreseeable future, the results will be more likely to be such a guide to than individual PVA models based on guessed-at or poorly estimated parameter values.

COMMENT 2.6

I find it difficult to pass judgement on the quality of the primary analysis, of sustained population declines. It is an interesting result, but it seems to me that there are many assumptions that need to be stated. There would be huge heterogeneity among species in this probability, but that is not considered here, and the way significant periods of decline is defined and estimated makes no accommodation for the imprecision inherent in estimation of declines.

RESPONSE: We disagree with the reviewer here. As we point out in the paper, the past performance of conservation scientists and ecologists in predicting which species will undergo sustained population declines is not impressive. Hence, while the probability of such declines may well be heterogenous, we do not think that it is practical, with present knowledge, to account for that heterogeneity in these analyses. The assumptions and methods of our analysis are outlined in great detail and Reviewer 2 makes very few detailed comments below on assumptions that we have neglected to specify. We therefore believe that we have addressed those comments. We may not have presented the assumptions concerning the accuracy of the original monitoring data used clearly enough. if this is where Reviewer 2’s criticism lies, we believe we have now addressed this. We note and appreciate that Reviewer 1 approves of the logic of our approach, and we have used these constructive comments of Reviewer 1 to clarify some points.

Specific comments:

COMMENT 2.7

Abstract: I find this to be quite confusing, as it pummels the readers with poorly described jargon. It could use some strong editing to provide a clear statement of the issue and how this analysis improves it. As I note elsewhere, it seems to me that the focus of the paper is the analysis of sustained declines, but the focus of the abstract is how the overall result ties into 1 component of FCS.

RESPONSE: Thank you. we have edited the abstract to clarify it and we have linked the abstract more closely to the focus of the paper.

COMMENT 2.8

L 28. I had not heard of the term Favorable Conservation Status, and this first sentence just does'nt make sense to me. This should be reworded to at last describe (even vaguely!) what FCS is, and perhaps how FRV-P (l. 30) relates to it!

RESPONSE: Favourable Conservation Status is concept much in use in the European Union. Favourable reference values FRVs are also such. We have removed them from the abstract and title and have defined ‘favourable’ as a term in the introduction in the context of other assessments such as those in the US and Canada. 

COMMENT 2.9

l. 55. Quantitative methods are always needed! I think you want to say that efficient methods are needed.

RESPONSE: We agree that it is good to add the word “efficient”. However, we did mean to say that the methods need to be quantitative as well. Previously (e.g. red-listing before the Mace-Lande criteria), conservation assessments of populations were efficient (in that they were done quite quickly and cheaply) but they were predominantly qualitative in most cases.

COMMENT 2.10

l. 59. You still have not defined this concept of "Favorable Conservation Status!" What does it mean? L 66-69 indicate that this concept relates to criteria we usually asspciate with risks of extinction, or species characteristics that provide a rationale for conservation activity. Please state this.

RESPONSE: As we pointed out in the Introduction, the main point of our paper is that FCS (an EU technical and legal concept) has NOT BEEN DEFINED satisfactorily in quantitative terms and we therefore aim to contribute a way to do that. For example, we stated clearly in the first paragraph of the Introduction “the term ‘favourable’ does not yet have a generally accepted definition”. We have redrafted the Introduction to present this and related concepts in a different way. We have removed the acronyms FCS and FRV-P but have kept the concept of what is a favourable population size which we believe (even if the reader is not familiar with the phrase Favourable Conservation Status) to be understandable to the reader.

COMMENT 2.11

l. 69 and following. You choose to focus on just one element of FCS. It seems to me that your introduction up to this point is misleading, as you are not discussing FCS comprehensively, but are just refining one element of it.

RESPONSE: We now focus on favourable population size as a measure and do not imply that the paper deals with the whole concept of FCS.

COMMENT 2.12

l. 71 and following. Pleae clarify the motivation for defining FRV-P. There seems to be a lot of semantics on this paragraph. You call this "numerical measures of population size." Is this the current population size (i.e, the population status), or is this a minimum viable population size? I find this confusing. Defining population status is common, and identifying species with small populations is often considered a risk factor for conservation assessments (i.e., Rabinowitz-style criteria), but these status values are much different than MPVs which relate specifically to a demographic modeling attribute. Statements in this paragraph suggest that you wish to define FRV-P as a modified MVP, but is that the actual intent of the FRV-P criterion?

RESPONSE: We think that the first two paragraphs of the Introduction already did this explicitly and clearly. However, we have responded by moving this part of the paper away from the EU concepts of FCS and FRV-P. We have rewritten the Introduction to clearly state what our motivations are and we hope this flows more clearly now. Our intention in the Introduction is still to say that favourable status, including population size, exists as a concept in legislation, but there is no accepted way of measuring or defining it. We give some examples where others have used methods that we do not think are satisfactory. We then state how our proposed method attempts to address some of the issues raised.

COMMENT 2.13

I believe that what you are saying here is the FRV-P is designed to be a MVP, and that your analysis provides an ad-hoc scaling of MVP to accommpdate stochastic effects not included in the original MVP formulation. If so, why not just define a more realistic MVP that includes such stockasticity?

RESPONSE: We think that the “more realistic MVP that includes such stochasticity” is not achievable for nearly all real animal populations. Only for a vanishingly small number of populations are (a) density dependence, (b) demographic rates, (c) environmental stochasticity’s effects on demographic rates known well enough and measured over a long enough period to build a reliable model of future risk. We contend that this is the case even for well-known species groups, such as birds. We therefore think that the reviewer’s proposal is unrealistic if the objective is to estimate long-term future risk for all or most of a large group of species. We have redrafted the section of the Introduction about MVPs to say this.

COMMENT 2.14

l. 90-93. This seems to me to be a odd comment. How do we know that several decades is short relative to the duration of some declines? It would be better to say that perhaps declines defined over short intervals are not predictive of long term declines and thus do not inform extinction probabilities (as Thogmartin and Stanton have done for North American birds).

RESPONSE: The reviewer has missed the point we are seeking to make here. Short-run monitoring populations does not reveal the full duration and magnitude of declines because their full probability distribution is veiled (i.e. short runs of data cannot tell you about long declines without special analysis). 

COMMENT 2.15

l. 121-123. This criterion defines a period of decline and recovery. the later criteria (up to line 135) are rules for finding the smallest n in the interval. Why not just say that?

RESPONSE: Our aim here was to define the rules for identifying SPDs explicitly in such a way that others could repeat it exactly. We don’t feel that the suggested replacement sentence achieves that unamabiguous clarity.

COMMENT 2.16

l. 138. You never state it explicitly, but presumably the duration of the SPD is from tstart to tlow.

RESPONSE: Later sections make it clear that defining the duration of SPDs requires that censoring is taken into account and we describe how that was done in detail. See lines 163-185.

COMMENT 2.17

SPDs must also be at least 10 years in duration. This seems very arbitrary, and likely to have a strong influence on the results.

RESPONSE: We accept that the ten-year threshold is arbitrary but do not think that our results are sensitive to this. Allowing a shorter threshold would inflate the number of SPDs detected but decrease the proportion of them that are high-magnitude. Hence, the effect on the multiplier calculation would be small.

COMMENT 2.18

l. 165 onward. All analyses after this point make identical distribution assumptions among the SBDs. Is that valid?

RESPONSE: We do not understand this point. “identical distribution assumptions”: identical with respect to what? We allow duration and magnitude to vary among species, but do not define subsets of species with differing distributions. That is because an empirical basis for doing that has not been established for this set of species, though it might be done in the future. Ignoring this possible subdivision for now does not seem to us to diminish the validity of our results. We have added text in the Discussion about what seems to us to be the most feasible method for making separate predictions of future SPD prevalence and magnitude for sub-groups of bird species. That is to do so using information about sensitivity to climatic change. However, the methods are not robust enough to do this yet.

COMMENT 2.19

l. 165-170. This seems to me to be an oversimplification. If so, why only consider d >9? Are there issues associated with not having d <10 in this analysis?

RESPONSE: We test the assumption of a constant annual probability of SPD cessation explicitly- see lines 179-182. So it is a simplification but one for which we provide justification. We address the point about the threshold of ten years in response to a point above.

---

## [Editor Report · Decision Letter 1]

23 Jan 2020

Implications of the prevalence and magnitude of sustained declines for determining a minimum threshold for favourable population size

PONE-D-19-27507R1

Dear Dr. Gilbert,

We are pleased to inform you that your manuscript has been judged scientifically suitable for publication and will be formally accepted for publication once it complies with all outstanding technical requirements.

With kind regards,

Floyd W Weckerly

Academic Editor

PLOS ONE
---

## [Editor Report · Acceptance letter]

28 Jan 2020

PONE-D-19-27507R1 

Implications of the prevalence and magnitude of sustained declines for determining a minimum threshold for favourable population size 

Dear Dr. Gilbert:

I am pleased to inform you that your manuscript has been deemed suitable for publication in PLOS ONE. Congratulations! Your manuscript is now with our production department. 

With kind regards,

on behalf of

Dr. Floyd W Weckerly 

Academic Editor

PLOS ONE